# Optimizing Light Flash Sequence Duration to Shift Human Circadian Phase

**DOI:** 10.3390/biology11121807

**Published:** 2022-12-13

**Authors:** Daniel S. Joyce, Manuel Spitschan, Jamie M. Zeitzer

**Affiliations:** 1Department of Psychiatry and Behavioral Sciences, Stanford University, Stanford, CA 94305, USA; 2Psychology Department, University of Nevada, Reno, NV 89557, USA; 3School of Psychology & Wellbeing, The University of Southern Queensland, Ipswich, QLD 4305, Australia; 4Department of Sport and Health Sciences, Technical University of Munich, 80992 Munich, Germany; 5Mental Illness Research Education and Clinical Center, VA Palo Alto Health Care System, Palo Alto, CA 94304, USA

**Keywords:** circadian rhythm, light, melatonin, sleep, dim light melatonin onset, flash

## Abstract

**Simple Summary:**

Information about light exposure patterns is integrated across time and space in retinal circuits before being passed to downstream hypothalamic targets, including the circadian pacemaker in the suprachiasmatic nuclei. In multiple mammalian species, these circuits can integrate brief flashes of light such that the effective impact on the circadian pacemaker is greater than that of continuous light of similar intensity. Using a series of 16-d studies combining outpatient behavioral manipulation and in-laboratory intensive physiologic monitoring, we examined the impact of different durations of light flash sequences on the timing of the human circadian pacemaker. We find that 15 min of light flashes engenders a similar effect on the timing of the human circadian pacemaker as light flashes given for 3.5 h. Our study indicates that the impact of a sequence of light flashes occurs within the first 15 min of exposure, and the retinohypothalamic circuit responsible for shifting the timing of the circadian pacemaker is refractory to further stimulation. These data are important for both the understanding of retinohypothalamic circuitry and the design of robust light interventions meant to change circadian timing during sleep, as would be used in the treatment of circadian rhythm sleep disorders.

**Abstract:**

Unlike light input for forming images, non-image-forming retinal pathways are optimized to convey information about the total light environment, integrating this information over time and space. In a variety of species, discontinuous light sequences (flashes) can be effective stimuli, notably impacting circadian entrainment. In this study, we examined the extent to which this temporal integration can occur. A group of healthy, young (*n* = 20) individuals took part in a series of 16-day protocols in which we examined the impact of different lengths of light flash sequences on circadian timing. We find a significant phase change of −0.70 h in response to flashes that did not differ by duration; a 15-min sequence could engender as much change in circadian timing as 3.5-h sequences. Acute suppression of melatonin was also observed during short (15-min) exposures, but not in exposures over one hour in length. Our data are consistent with the theory that responses to light flashes are mediated by the extrinsic, rod/cone pathway, and saturate the response of this pathway within 15 min. Further excitation leads to no greater change in circadian timing and an inability to acutely suppress melatonin, indicating that this pathway may be in a refractory state following this brief light stimulation.

## 1. Introduction

The non-image forming (NIF) retinal pathways signal information about environmental illumination to diverse brain targets, including the suprachiasmatic nucleus (SCN), which regulates circadian physiology. The origin of the NIF visual pathways is the melanopsin-expressing intrinsically photosensitive retinal ganglion cells (ipRGCs) located in the inner retina. These ipRGCs combine inputs from the temporally precise rod and cone photoreceptors (extrinsic pathway) with temporally sluggish signaling from the melanopsin photopigment (intrinsic pathway) to signal afferent brain targets [1]. The functional significance of these combined fast and slow pathways is not yet fully elucidated for NIF vision, but may confer some advantages, such as setting retinal gain and optic nerve firing rates [2,3], optimizing pupil size in dynamic environments [4,5], or integrating color and intensity information for setting circadian rhythms [6].

The SCN receives signals from both of these fast and slow pathways [7], suggesting a unique contribution of temporally sluggish and temporally precise photic information that, together, modulate the circadian pacemaker and emergent sleep–wake behavior. In support of this, a growing body of evidence shows that the human circadian pacemaker is extremely sensitive to the temporal patterning of light stimuli. Sequenced flashes of light presented during the biological night (~470 lux broad spectrum light, 2-millisecond flashes every 60 s for an hour) delay the circadian phase by 45 min [8]. This delay was equivalent to that from ~50 lux of light, some one-tenth of the light intensity, but was presented continuously for 6.5 h, which is ~200,000 times the total duration [9]. Further optimization of the flash paradigm revealed a strong nonlinear dependence on the interstimulus interval (ISI) between flashes. Bright flashes (~1700 lux broad spectrum light, 2-millisecond flashes for an hour) yielded a maximum drive when separated in time by 7.6 s, and resulted in phase delays of up to 3 h in some participants [10].

It is unknown how circadian responses depend upon the total duration of the flashed light sequence. A previous examination of different durations of continuous light exposures in humans (0.2 h, 1.0 h, 2.5 h, 4.0 h) indicated a semi-logarithmic relationship between duration of light exposure and the resulting change in circadian phase [11]. The results, however, did not effectively predict the response to very short, i.e., 15-s or 2-min, duration exposure. Very short exposure to continuous light [12] or a series of millisecond-long light flashes [10] also elicits more robust circadian phase shifting than would be anticipated from the responses to long-duration exposure to light. It is unknown whether these responses to very short light exposures use a different retinal mechanism to transduce light information into a neurochemical signal. Flashed light sequences, at the very least, are likely to have a different ratio of extrinsic to intrinsic signaling to the SCN as compared to continuous light, with an expected greater (but intermittent) drive from the outer retinal pathway that could, at least partially, re-sensitize during the periods of darkness between flashes. It is unknown if this re-sensitization decays over time. The purpose of this study was to determine the dose-response relationship between the paradigm duration of exposure to a sequence of millisecond-long light flashes, and the resulting circadian phase shift.

## 2. Materials and Methods

### 2.1. Participants

We recruited 20 (7 female) healthy participants to take part in this study (Figure 1), aged 21 to 35 (28.6 ± 4.86 years). Inclusion criteria included no sleep disturbances (Pittsburgh Sleep Quality Index score ≤5; [13]), no alcohol abuse (Alcohol Use Disorders Identification Test score ≤5; [14]), and no depression (Center for Epidemiologic Studies–Depression score ≤19; [15]). Participants had an intermediate chronotype (Morning–Eveningness Questionnaire (simplified version) scores 11–26; [16]) and normal color vision [17]. Female participants attended the lab within four days of the onset of menses.

### 2.2. Procedures

#### 2.2.1. Protocol

Participants took part in a 16-day protocol that consisted of 14 days of sleep–wake monitoring in the home environment, followed by 35 to 36 h of in-laboratory testing (Figure 2). For Days 1 to 14 at home, participants maintained a regular sleep–wake rhythm (sleep and wake times within ±30 min of target times, set 7–9 h apart) with self-selected sleep and wake times. Compliance was monitored via actigraphy (Philips Actiwatch 2, Bend OR or Ambulatory Monitoring Motionlogger, Ardsley, NY, USA) and a modified version of the Consensus Sleep Diary [19]. On Day 15, participants came to the laboratory in the afternoon. Participants who tested positive for recreational drug (IDTC-II 6 Panel Instant Drug Test Card, CLIAwaived, San Diego, CA USA) or alcohol (ALCO Screen 02, Chematics, North Webster, IN, USA) use, or whose sleep or wake times deviated by more than ±30 min twice across the 14 days, were disempaneled. All events were scheduled according to the midsleep phase (MSP), calculated as the average of the at-home sleep midpoints determined through examination of actigraphy and sleep diary data.

The laboratory study took place in a custom light-controlled and time isolation suite with an *en suite* bathroom at the Veterans Affairs Palo Alto Health Care System, from the evening of Day 15 until the conclusion of the study on Day 16. To control and minimize the effects of light on the circadian system during waking hours, dim room lighting (0.6–1.9 lux, measured in the horizontal angle of gaze) was maintained by ceiling-recessed dimmable fluorescent lamps (GE Covrguard 4000 °K, East Cleveland, OH, USA). During scheduled sleep periods, all lighting was turned off (<0.05 lux). 

During the first evening (Day 15), participants underwent a constant posture (CP) procedure to hold constant or remove factors that might otherwise mask endogenous circadian rhythms [20]. During the CP, participants maintained a semi-recumbent position in a hospital bed and were given hourly isocaloric snacks (Ensure Plus, Abbott Laboratories, Lake Bluff, IL, USA) provided in lieu of dinner, adjusted to metabolic demands [21]. After adapting to the CP for an hour, saliva samples were collected every 30 min by expectorating into an untreated polypropylene tube. All saliva samples were immediately placed on ice and then transferred to a −80 °C freezer until assay (see below). Alertness and sleepiness levels were monitored every 60 min (see below). Participants were allowed use of a bedpan or urine bottle as needed. This first CP ended 4 h before MSP, at which time the room lighting was extinguished and participants were asked to sleep.

Participants were awakened into darkness by a laboratory member at the appropriate time (see Light stimulation and timing) and fitted with a custom-made eye mask to deliver the light sequences. Saliva was sampled immediately before flashes commenced and either 10 min into the protocol (0.25-h condition, 2 samples), every 20 min (1-h condition, 4 samples), or every 30 min (2-h and 3.5-h conditions, 5 and 8 samples, respectively). Sleepiness and alertness were assessed immediately before the light stimulus, prior to the end of the light stimulus for the 0.25-h and 1-h conditions, and approximately hourly for the 2-h and 3.5-h conditions. Following this experimental light exposure, participants were encouraged to return to sleep in darkness until being awakened at their habitual wake time 4 h after MSP (Day 16). On Day 16, participants were free to move around the dimly lit time isolation suite and use the *en suite* bathroom; a standard hospital breakfast and lunch were provided. On the evening of Day 16, participants engaged in a second CP before being discharged at the conclusion of the study.

#### 2.2.2. Light Stimulation and Timing

To deliver the specified light flash sequence and duration, participants were fitted with custom full-field hemispheric diffusers over each eye (Stiga Perform 40 + white, Eskilstuna, Sweden). Over the diffusers, participants wore a custom-made eye mask (Jackson Safety 15991, Elgin IL, USA) fitted with 4000 K LEDs (Lumileds LUXEON XF-3535L, San Jose, CA, USA) that projected through an acrylic diffusing panel (TAP Plastics, Mountain View, CA, USA). The spectral distribution of the light [22] was intended to activate any of the ocular photoreceptive systems (i.e., rods, cones, melanopsin). The timing and intensity of light was controlled by a 16 MHz programmable microcontroller (Arduino Uno R3, Turin, Italy). Stimuli consisted of 2 ms flashes of 2200-lux light presented every 8 s; the intensity of the light was confirmed prior to the study by a calibrated photometer (International Light Technologies ILT900, Peabody, MA, USA). The stimulus was viewed for durations of either 0.25, 1, 2, or 3.5 h, depending on randomized allocation; all were presented during the phase delay portion of the phase response curve to light [23]. Five participants were allocated into each condition; we removed the data of one participant (1-h exposure) due to failure of the lighting apparatus during the study. Each stimulus duration was aligned such that the midpoint of each presentation duration occurred at the same time relative to the MSP for each participant, 1.5 h prior to the MSP (i.e., 2.5 h after habitual sleep time). Participants, therefore, had a sleep opportunity that ranged from 0.5 h (3.5-h protocol) to 2.125 h (0.25-h protocol) prior to being awakened for the light exposure.

#### 2.2.3. Assessment of Melatonin

Saliva samples were assayed for melatonin concentrations using an enzyme-linked immunosorbent assay (salivary melatonin ELISA #3402, Salimetrics, Carlsbad, CA, USA; assay range: 0.78–50 pg/mL, sensitivity = 1.37 pg/mL). The onset of salivary melatonin on Day 15 (baseline) and Day 16 (post-stimulus) was determined as the time at which salivary melatonin concentrations exceeded twice the average of the concentration of the first three samples for each respective day [24]. In four participants, we set the threshold at 10 pg/mL due to noisy baselines. All determinations of melatonin phase assessments were conducted blind, in randomized conditions. The phase angle (ψ) of light exposure was calculated post hoc as the time between the onset of melatonin and the onset of light exposure. The phase change of melatonin (Δϕ) was, by convention, calculated as the difference in the time of melatonin onset on Day 15 and Day 16, causing delays in the timing to be negative. Acute suppression of melatonin was calculated as the percent change in melatonin concentration between the sample immediately preceding the light sequence and the sample obtained immediately prior to the end of the light sequence. Negative values indicate a reduction in, or ‘suppression’ of, melatonin concentrations.

#### 2.2.4. Assessment of Alertness and Sleepiness

Subjective sleepiness was assessed using the Stanford Sleepiness Scale (SSS), a 1-item, 7-point Likert-like scale that probes current sleepiness, with higher scores indicating greater sleepiness [25]. The SSS was administered verbally to the participant. So as not to introduce untoward ambient light, objective alertness was measured with a custom-built auditory version of the Psychomotor Vigilance Task (aPVT) [26,27]. The device was based on an Arduino Uno microcontroller board interfaced via Python. Participants were instructed to press a button as quickly as possible in response to perceiving a 1000 Hz tone. Each tone was preceded by a randomized interval of 1 to 6 s [28], and continued until the button was pressed, with no feedback provided to the participant. Each session lasted 10 min and comprised ~100 trials. The percent of trials that ended in a lapse, the median reaction time during the trial, and the average of the fastest 10% of the reaction times were selected a priori as outcome measures of the aPVT.

#### 2.2.5. Statistics

All statistical analyses were performed in Origin Pro 8.0891 (OriginLab Corp., Northampton, MA, USA), and are indicated in the text. Data are presented as mean ± SD unless otherwise noted.

## 3. Results

### 3.1. Melatonin Phase Shifts

Participants received light centered 4.5 ± 0.94 h after the onset of melatonin. The responses to a flash sequence lasting 15 min (−0.76 ± 0.72 h), 1 h (−0.65 ± 0.28 h), 2 h (−0.48 ± 0.43 h), or 3.5 h (−0.72 ± 0.89 h) were each statistically distinguishable from a dark protocol control condition [8], in which a shift of 0.11 ± 0.25 h was observed (*p*’s < 0.05, Mann–Whitney tests) (Figure 3A). The responses to exposure of different durations were not, however, distinguishable from one another (*p* = 0.58, Χ^2^ = 1.96, Kruskal–Wallis ANOVA).

### 3.2. Melatonin Suppression

Following a 1-h (8.45 ± 26.1% increase), 2-h (12.0 ± 34.2% decrease), and 3.5-h (30.1 ± 36.1% increase) exposure to a light flash sequence, there were no differences in melatonin concentrations when comparing the sample just before the light sequence to the sample at the end of the light sequence (*p*’s > 0.5, Wilcoxon Signed Ranks Tests) (Figure 3B). Following a 15-min exposure to a light flash sequence, there was a 17.8 ± 2.32% decline in melatonin concentrations (*p* < 0.05, Wilcoxon Signed Ranks Test) (Figure 3B). These acute changes in melatonin concentrations were distinguishable from one another (*p* > 0.05, Χ^2^ = 8.02, Kruskal–Wallis, ANOVA) with the decrease in melatonin following 15 min of exposure being different from the increase in melatonin following 3.5 h of exposure (*p* < 0.05, post hoc Dunn’s Test).

### 3.3. Subjective Alertness

Upon being aroused in the night, participants rated themselves as 4.8 ± 1.4 on the SSS. Following a 1-h (0.50 ± 0.58 point decline), 2-h (1.0 ± 2.2 point decline), and 3.5-h (2.4 ± 2.7 point increase) exposure to a light flash sequence, there were no differences in SSS scores when comparing the test just before the light sequence to the test at the end of the light sequence (*p*’s > 0.17, Wilcoxon Signed Ranks tests). Following a 15-min exposure to a light flash sequence, there was a 1.00 ± 0.71 point decline in SSS scores (*p* < 0.05, Wilcoxon Signed Ranks tests). These changes in SSS scores were not different among the four exposure durations (*p* = 0.17, Χ^2^ = 5.02, Kruskal–Wallis, ANOVA).

### 3.4. Objective Alertness

Upon being aroused in the night, across all participants, 2.5 ± 2.8% of the trials on the aPVT were lapses, the median response time was 411 ± 140 ms, and the average of the fastest 10% of response times was 307 ± 88.2 ms (Table 1). There were no significant changes in any of the metrics in any of the conditions from before to after the stimulus (*p*’s > 0.14, paired t-tests). The changes in aPVT in response to the flash sequences did not differentiate by duration for the percent of trials with a lapse (*p* = 0.15, Χ^2^ = 5.26, Kruskal–Wallis, ANOVA), median response time (*p* = 0.59, Χ^2^ = 1.93, Kruskal–Wallis, ANOVA), or the fastest 10% of response times (*p* = 0.74, Χ^2^ = 1.25, Kruskal–Wallis, ANOVA).

## 4. Discussion

ipRGCs form the neural substrate for circadian phototransduction. They directly transduce visible electromagenetic radiation via the photopigment melanopsin while also integrating signals originating from rod and cone photoreceptors in the outer retina [29]. Here, we attempted to bias activation towards the outer retinal inputs with flashed light stimuli (2-ms flashes and 8-s ISIs) that preferentially activate cones [30]. In doing so, we found that exposure to flash sequences longer than 15 min does not evoke any greater shift of the human circadian clock. The subtle melatonin suppression observed during a 15-min sequence of flashes, which was not accompanied by objective changes in alertness, was also unable to be maintained beyond the 15-min exposure. Our data are consistent, but not exclusionary, with the hypothesis that cones are only able to contribute to short exposures to light and that sustained exposures are unable to maintain outer retinal input to the ipRGC to SCN circuit. This is also consistent with data obtained from continuous light exposure paradigms, in which the beginning of the light exposure appears to be driven more by cones than by melanopsin [31], and light signals transduced by rods/cones capably modulate circadian behaviors with melanopsin knockout [32]. Further, continuous light exposures, likely mediated mostly by melanopsin [33], presented at the same circadian phase, but with 4.5-fold greater intensity (~10,000 lux), engenders similar magnitude phase delays (~40 min) for 12-min exposures, but larger phase delays for exposures of 1 h (~1.1 h), 2 h (~1.8 h), and 4 h (~2.2 h) [11].

The physical parameters of flashed light sequences—flash duration, flash intensity, interflash interval duration, and total sequence duration—have specific impacts on the human phase shifting responses to such sequences. Flash durations from 10–1000 µs yield similar phase shifts [22], as do duration exposures 15–210 min (present study) and flash intensities 30–9500 lux [22]; the interflash interval has a non-monotonic (rapid peak followed by exponential decay) dose-response relationship [10]. These data follow observations that outer retinal mechanisms provide a functionally significant ON signal, with a fast decay [34] to higher-order circadian centers that augment, but are dissociable from, signals originating via melanopsin phototransduction [7].

Multiple pathways confer the circadian system with temporal and intensity resolution to sense the full gamut of environmental lighting. Mouse ex vivo analysis revealed that retinal photoreceptors are required to encode flashes, and are not diminished by knocking out melanopsin [30]. Sleep induction of mice by flashes is reduced, but not obviated by melanopsin’s absence [35]. The multiplexing of light information through mechanisms of different sensitivity may optimally identify lighting changes that are environmentally significant, such as the timing of dawn and dusk [30].

While our data are consistent with the theory that cone photoreceptors are involved in the transduction of a sequence of flashes to the circadian clock, our findings do not obviate the possibility that rods or melanopsin are involved. When light is administered continuously, melanopsin is the main transducer of the energy to the circadian system [33,36]. While the spectral content of the light stimulus used in this study was broad enough to engender responses from melanopsin, the peak of which was at ~480 nm, the magnitude of the responses and the temporal characteristics of the light flashes reduced the likelihood of the involvement of melanopsin. This indicates that there may be a unique role for cone photoreception in the resetting of the clock, at least to a sequence of light flashes. The role that cone photoreception has in synchronizing the circadian clock to a more traditional light diet is still not well-understood. Further studies on individuals lacking outer cone segments would be needed to fully resolve this issue. Additionally, studies that expand the number of participants could resolve subtle differences in the different durations, but these data indicate that it is unlikely that such differences would be of large magnitude.

## 5. Conclusions

A 15-min sequence of light flashes generates as much drive on the human circadian clock as does a 3.5-h sequence, which is consistent with the hypothesis that most of the drive generated by light flashes occurs at the initiation of exposure. Our data are consistent with the theory that cone photoreceptors are responsible for encoding light flash sequence energy to the circadian clock. The data also imply that following an initial responsivity to the flashes, there is a refractory period during which the retinohypothalamic circuit either no longer encodes or is no longer responsive to additional stimuli.

## Figures and Tables

**Figure 1 biology-11-01807-f001:**
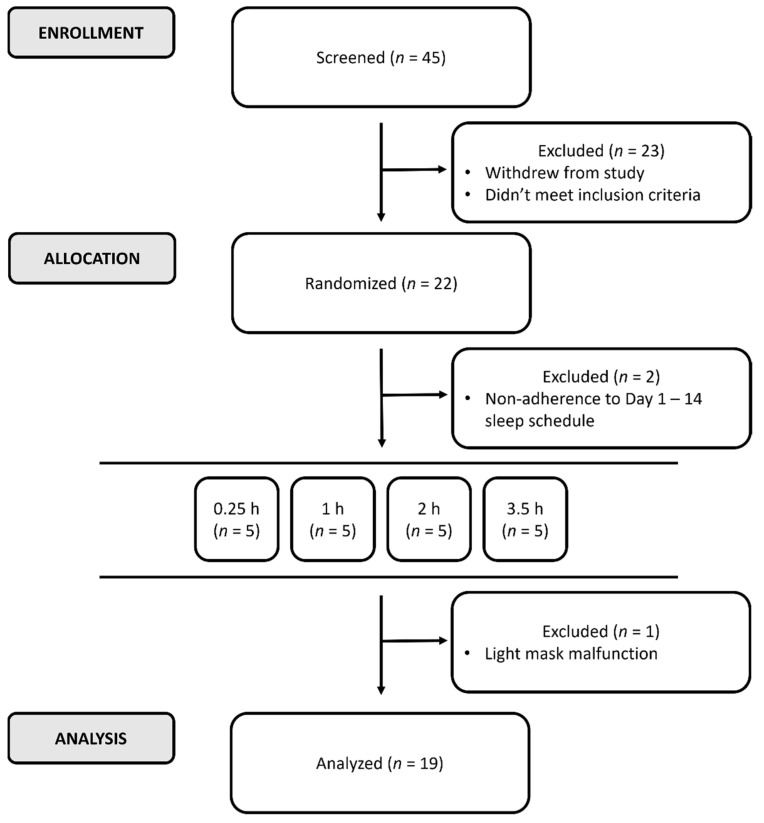
CONSORT diagram [18] of participant recruitment and allocation. Of the individuals evaluated, 51% did not meet inclusion criteria or declined to take part in the experiment. Of the participants empaneled, 9.1% were excluded due to non-adherence to target at-home sleep/wake times. Twenty-two participants took part in the in-lab component, quasi-randomized into five participants per illuminance level (see Protocol). The data of one participant (5%) was excluded in the 1 h condition due to failure of the experimental device.

**Figure 2 biology-11-01807-f002:**
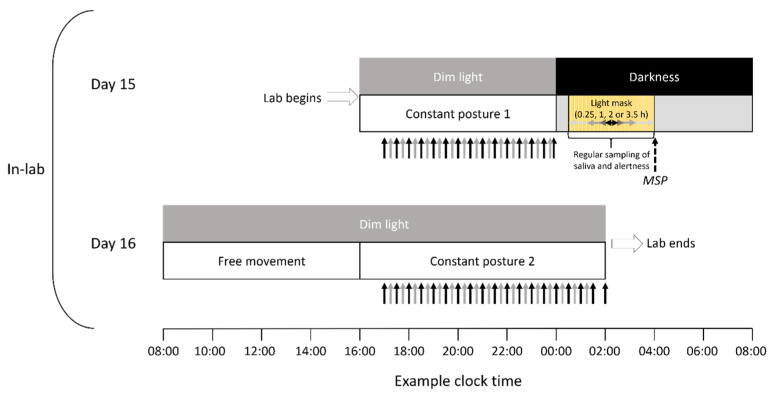
Timing of the laboratory protocol. After stabilizing their sleep/wake rhythms for fourteen days, participants attended the laboratory for a controlled study of the effect of stimulus duration on circadian rhythms. Upon admission to the laboratory on Day 15, participants remained in dim light (<2 lux) during constant posture, regularly provided saliva samples (solid black arrows), and were tested for objective and subjective sleepiness (solid gray arrows). Room lighting was extinguished 4 h prior to the midsleep phase (MSP, dashed black arrow) to provide a sleep opportunity. During the biological night, participants were awakened and exposed to a sequence of light flashes via a light-emitting eye mask. Light exposure lasted for either 0.25, 1, 2, or 3.5 h, and consisted of 2-millisecond flashes of ~2200 lux white light, which were presented every 8 s. The stimuli were aligned such that their midpoints occurred at the same circadian time. On Day 16, participants were awakened at their habitual wake time under dim light, and were provided breakfast and lunch before entering constant posture to, again, provide measures of saliva and of objective and subjective alertness. Participants were discharged at the conclusion of constant posture 2.

**Figure 3 biology-11-01807-f003:**
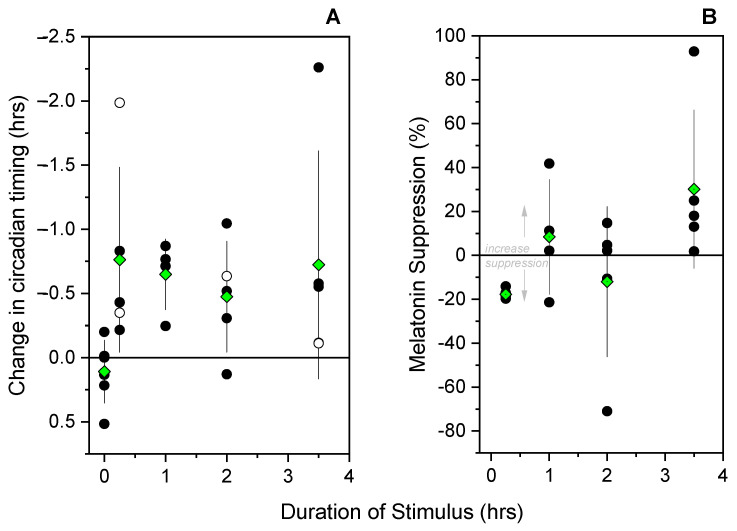
Phase shifting in response to light flash sequences of different durations. Left panel (**A**): The magnitude of the shift in the timing of the onset (phase) of melatonin (Δϕ) is plotted against the duration of the flash sequence and compared to a historic protocol control (8) in which individuals were awakened into darkness and not given a light stimulus (0 h data). Right panel (**B**): The magnitude of the change in acute melatonin concentrations between the beginning and the end of the flash stimulus. Individual data (black discs) are displayed along with group average (green diamonds) with SD bars. The open discs indicate the four participants in whom a fixed, rather than dynamic, threshold was used to determine circadian timing.

**Table 1 biology-11-01807-t001:** Performance metrics derived from an aPVT administered immediately before and at the end of a sequence of light flashes lasting between 15 min and 3.5 h. The percentage of trials that were defined as lapses, median reaction time, and average reaction time of the fastest 10% of the trials are presented.

	% Lapses		Median Reaction Time		Fastest 10%	
	Pre	Post	Pre	Post	Pre	Post
15-min	2.56 ± 2.46%	1.04 ± 1.48%	440 ± 127 ms	386 ± 75.7 ms	315 ± 52.8 ms	289 ± 53.2 ms
1-h	5.28 ± 4.20%	2.40 ± 2.91%	351 ± 30.7 ms	343 ± 13.2 ms	265 ± 15.7 ms	266 ± 17.8 ms
2-h	2.27 ± 1.68%	2.93 ± 2.23%	336 ± 61.8 ms	333 ± 95.4 ms	265 ± 41.3 ms	262 ± 65.8 ms
3.5-h	0.541 ± 1.21%	2.09 ± 2.22%	504 ± 213 ms	420 ± 13.6 ms	373 ± 145 ms	321 ± 32.5 ms

## Data Availability

The data that support the findings of this study are available from the Dryad Digital Repository: https://doi.org/10.5061/dryad.xd2547dkc.

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
