# Peer review of "Optimizing Light Flash Sequence Duration to Shift Human Circadian Phase"

_biology, 2022, doi:10.3390/biology11121807_

Round 1

Reviewer 1 Report

The authors of the paper "Optimizing light flash sequence duration to shift human circadian phase" tackled a very interesting topic and tested the effectiveness of short-duration light flashes on the timing of melatonin rise and suppression and alertness. Although I completely agree with the reasons why these studies are necessary and important, which the authors explain for example at the end of "Simple Summary", I do not really like the way the melatonin data are presented.

First of all, if I understand the methodology correctly, there are 16 participants who were repeatedly tested before and after the pulses. I think a pairwise comparison would be appropriate, at least for Figure 3A. In my opinion, this representation is on the one hand methodologically cleaner and on the other hand, it also allows for the assessment of inter-individual differences and their discussion, for example in the context of Phillips 2019 (www.pnas.org/cgi/doi/10.1073/pnas.1901824116). In addition, 4 participants whose DLMO was calculated differently from the rest disappeared in this chart, and this should also be visible. Personally, I appreciate it when whole graphs are available for inspection, at least as supplements. For a less experienced reader, it can be difficult to imagine what the DLMO calculation stands for (Biology is not a field-specific journal). Also confusing is the designation of the "y" axis in 3A, which does not quite match the methodological description of the calculation of ΔΦ.

Another thing - looking at Figure 2, I have the impression that the sampling on day 16 was done in "dim light" still at the time when the sampling on day 15 was already done in complete darkness. But this cannot be fully compared. The theory that dim light intensity has no effect on melatonin,  is refuted for example, by the above-mentioned study. If this is not the case and both samplings took place under comparable lighting conditions, please try redesigning Fig. 2 so that this is not the first thought that comes to the reader's mind.

A question of interest - is there a model of the dynamic responses of the cones on which the timing of the light pulses in this experimental setup is based?

Author Response

First of all, if I understand the methodology correctly, there are 16 participants who were repeatedly tested before and after the pulses. I think a pairwise comparison would be appropriate, at least for Figure 3A. In my opinion, this representation is on the one hand methodologically cleaner and on the other hand, it also allows for the assessment of inter-individual differences and their discussion, for example in the context of Phillips 2019 (www.pnas.org/cgi/doi/10.1073/pnas.1901824116).

Unfortunately, the participants were allocated to a single pulse length each (see figure 1). So, while we completely agree that a paired statistical approach would have been stronger, we are unable to do that for these data.

In addition, 4 participants whose DLMO was calculated differently from the rest disappeared in this chart, and this should also be visible.

We have modified the figure to indicate the four participants in whom DLMO was calculated differently.

Personally, I appreciate it when whole graphs are available for inspection, at least as supplements.

Due to restrictions of one of the employers (U.S. Department of Veterans Affairs), the individual data plots cannot be made publicly available as a supplement. These data, however, are obtainable by request.

For a less experienced reader, it can be difficult to imagine what the DLMO calculation stands for (Biology is not a field-specific journal). Also confusing is the designation of the "y" axis in 3A, which does not quite match the methodological description of the calculation of ΔΦ.

We have added verbiage to the Figure 3 legend to better match the description in the Methods.

Another thing - looking at Figure 2, I have the impression that the sampling on day 16 was done in "dim light" still at the time when the sampling on day 15 was already done in complete darkness. But this cannot be fully compared. The theory that dim light intensity has no effect on melatonin, is refuted for example, by the above-mentioned study. If this is not the case and both samplings took place under comparable lighting conditions, please try redesigning Fig. 2 so that this is not the first thought that comes to the reader's mind.

The sampling during the constant posture was done in dim light on both days. The sampling on day 15 that was done during the light pulse was done in darkness (except for the experimental light).

 A question of interest - is there a model of the dynamic responses of the cones on which the timing of the light pulses in this experimental setup is based?

We originally designed this protocol based on the previous study of Chang et al. 2012 that examined the duration-dependence of circadian responses to continuous light exposure. We had not anticipated the duration invariance and, as you suggest, altering timing on a dynamic cone model would have made more sense in retrospect.

Reviewer 2 Report

Thank you very much for the great work. It was a good read! You have kept a high scientific standard.

Congratulation.

Minor

Line 60. The term broadband light is widely associated with therapies for human skin. The electromagnetic radiation used here is harmful to the eye, therefore I suggest using another term, such as high quality broad spectrum light.

Major

Line 85. Please address the human wavelength-sensitive melanopic light effect, as the topic seems important to the reader in this context. The wavelength of the light to which the subject is exposed has a significant influence on the biological effect of light including the shift of the circadian rhythm mentioned in the text. Eventually, melatonin suppression is among the main statements later on. I recommend that you include the following sources in the document for this purpose. Only then the reader will be able to recognize the importance of the color temperature of the illuminants used in the experiment and described by you later one.

Here is a relevant source

Lucas, R.J.; Peirson, S.N.; Berson, D.M.; Brown, T.M.; Cooper, H.M.; Czeisler, C.A.; Figueiro, M.G.; Gamlin, P.D.; Lockley, S.W.; O’Hagan, J.B.; et al. Measuring and using light in the melanopsin age. Trends Neurosci. 2014, 37, 1–9.

Here is a suitable standard

International Commission on Illumination. CIE TN 003:2015, Report on the First International Workshop on Circadian and Neurophysiological Photometry; International Commission on Illumination: Vienna, Austria, 2015.

And here is a matching overview study

Neberich, M.; Opferkuch, F. Standardizing Melanopic Effects of Ocular Light for Ecological Lighting Design of Nonresidential Buildings—An Overview of Current Legislation and Accompanying Scientific Studies. Sustainability 2021, 13, 5131. https://doi.org/10.3390/su13095131

Minor

Line 95. At this point in the text, the reader does not yet know the meaning of the intervention group labels (0.25 h – 3.50 h). Perhaps better exchange this paragraph with the paragraph Procedure.

Line 120. Please increase the readability of this graphic by embedding it in a lossless file format.

Line 173. The maximum sensitivity of the human melatonin level to ambient light is achieved with light of a wavelength of more or less 490 nm, please describe to what extent you achieve the sensitivity with your illuminant in the experiment (4,000 K). Please tell the reader the reasons for the selection of the illuminant.

Line 174. Kelvin without “°“

Line 227. At this point in the text, it is not yet clear how exactly the time difference is calculated. Please insert a sentence or two for clarification. Additionally, a small table with the following data would be helpfull (not a must).

Major

I could imagine that it is possible to get more out of the results with further statistical methods. You can plot the p-values in a table and calculate the RTEs. All in all, I think the work is solid, but it seems to me that you can extract more interesting information to take the publication to an even higher level. For example, I think it is interesting that the circadian time shift does not increase linearly with exposure time. For this context, a diagram would be helpful to understand.

As soon as further findings are added, it would also be helpful to briefly summarize everything in a Conclusion.

Br

Author Response

Line 60. The term broadband light is widely associated with therapies for human skin. The electromagnetic radiation used here is harmful to the eye, therefore I suggest using another term, such as high quality broad spectrum light.

We have changed “broadband” to “broad spectrum”

Line 85. Please address the human wavelength-sensitive melanopic light effect, as the topic seems important to the reader in this context. The wavelength of the light to which the subject is exposed has a significant influence on the biological effect of light including the shift of the circadian rhythm mentioned in the text. Eventually, melatonin suppression is among the main statements later on. I recommend that you include the following sources in the document for this purpose. Only then the reader will be able to recognize the importance of the color temperature of the illuminants used in the experiment and described by you later one. Here is a relevant source: Lucas, R.J.; Peirson, S.N.; Berson, D.M.; Brown, T.M.; Cooper, H.M.; Czeisler, C.A.; Figueiro, M.G.; Gamlin, P.D.; Lockley, S.W.; O’Hagan, J.B.; et al. Measuring and using light in the melanopsin age. Trends Neurosci. 2014, 37, 1–9. Here is a suitable standard: International Commission on Illumination. CIE TN 003:2015, Report on the First International Workshop on Circadian and Neurophysiological Photometry; International Commission on Illumination: Vienna, Austria, 2015. And here is a matching overview study: Neberich, M.; Opferkuch, F. Standardizing Melanopic Effects of Ocular Light for Ecological Lighting Design of Nonresidential Buildings—An Overview of Current Legislation and Accompanying Scientific Studies. Sustainability 2021, 13, 5131. https://doi.org/10.3390/su13095131

While melanopsin has been established as a primary contributor to continuous light, it is unknown whether it has any role in mediating responses to flash sequences. All of our exposures have a constant melanopic content, so this does not change the interpretation of the results. Our prior work indicates that it is unlikely that melanopsin plays a significant role in responses to flashed light.

Line 95. At this point in the text, the reader does not yet know the meaning of the intervention group labels (0.25 h – 3.50 h). Perhaps better exchange this paragraph with the paragraph Procedure.

We have added a note in the Figure legend to see that specific section.

Line 120. Please increase the readability of this graphic by embedding it in a lossless file format.

This will be done in the final product (limited by journal requirements).

Line 173. The maximum sensitivity of the human melatonin level to ambient light is achieved with light of a wavelength of more or less 490 nm, please describe to what extent you achieve the sensitivity with your illuminant in the experiment (4,000 K). Please tell the reader the reasons for the selection of the illuminant.

The 4000 K color temperature was selected as it provided a broad spectrum of light that could impact melanopsin, rods, and cones. It was not selected to preferentially activate any single photoreceptive system. We have added verbiage to this extent.

Line 174. Kelvin without “°“

Done.

Line 227. At this point in the text, it is not yet clear how exactly the time difference is calculated. Please insert a sentence or two for clarification. Additionally, a small table with the following data would be helpfull (not a must). The difference calculation is presented earlier in the Methods (202-204).

I could imagine that it is possible to get more out of the results with further statistical methods. You can plot the p-values in a table and calculate the RTEs. All in all, I think the work is solid, but it seems to me that you can extract more interesting information to take the publication to an even higher level. For example, I think it is interesting that the circadian time shift does not increase linearly with exposure time. For this context, a diagram would be helpful to understand.

We had done extensive curve fitting with these data, but none of the models (notably: linear, 3 and 4 parameter log, logistic, logarithmic) had adequate fits.

As soon as further findings are added, it would also be helpful to briefly summarize everything in a Conclusion.

We have added an additional summary at the end of the Discussion.

Reviewer 3 Report

Hypothesis: To investigate the dose-response relationship between duration of millisecond light flash exposure and resulting circadian phase shift.

19 subjects (there are some places where this is muddled b/t 19 and 20).  

Demographics regarding subjects must be included.  Also, results from actigraphy (individual plots of phase, mesor, etc) would help better describe baseline data.  Furthermore, any differences in phase calculations done using actigraphy/sleep profiling and melatonin sampling would be beneficial to the community.

Additional data should be provided to confirm findings as significant.  Graphs describe changes in phase and melatonin output, why not show baseline phase/mel and draw lines connecting individuals?  This may uncover intra-subject variability that might be due to phasing. Eg, people who are delayed might be respond differently to acute light exposure than those who are advanced.

Author Response

Demographics regarding subjects must be included.  

Demographics are included in the Methods, lines 88-94.

Also, results from actigraphy (individual plots of phase, mesor, etc) would help better describe baseline data.  Furthermore, any differences in phase calculations done using actigraphy/sleep profiling and melatonin sampling would be beneficial to the community.

As the phase of sleep was fixed by the protocol, we unfortunately could not assess the impact of the intervention on the timing of the next night of sleep. We also could not examine changes in the polysomnographic profile of sleep on the next night of sleep as this subsequent sleep was not in the laboratory and followed variable environmental light exposure.

Graphs describe changes in phase and melatonin output, why not show baseline phase/mel and draw lines connecting individuals?  This may uncover intra-subject variability that might be due to phasing. Eg, people who are delayed might be respond differently to acute light exposure than those who are advanced.

We attempted fitting many different models (linear, 3 and 4 parameter log, logistic, logarithmic), but none were significant. As such, we chose not to connect the different groups. Individuals plots will be made available upon request, but cannot be published due to privacy restrictions from one of the author’s employers (US Department of Veterans Affairs.

Round 2

Reviewer 1 Report

The authors have sufficiently explained their points of view in their answers. I think the manuscript can be accepted

Author Response

Thank you for your helpful commentary!

Reviewer 2 Report

To be honest, I was now disappointed when I saw your very few corrections. Take the chance and increase the interest level of your publication before release.

You didn't include any other tables or diagrams that I asked for, and you only added a few sentences or sentence fragments. All in all most of the points I raised were either not addressed, or were addressed to an insufficient extent.

Line 85. Please address the human wavelength-sensitive melanopic light effect, as the topic seems important to the reader in this context. The wavelength of the light to which the subject is exposed has a significant influence on the biological effect of light including the shift of the circadian rhythm mentioned in the text. Eventually, melatonin suppression is among the main statements later on. I recommend that you include the following sources in the document for this purpose. Only then the reader will be able to recognize the importance of the color temperature of the illuminants used in the experiment and described by you later one. Here is a relevant source: Lucas, R.J.; Peirson, S.N.; Berson, D.M.; Brown, T.M.; Cooper, H.M.; Czeisler, C.A.; Figueiro, M.G.; Gamlin, P.D.; Lockley, S.W.; O’Hagan, J.B.; et al. Measuring and using light in the melanopsin age. Trends Neurosci. 2014, 37, 1–9. Here is a suitable standard: International Commission on Illumination. CIE TN 003:2015, Report on the First International Workshop on Circadian and Neurophysiological Photometry; International Commission on Illumination: Vienna, Austria, 2015. And here is a matching overview study: Neberich, M.; Opferkuch, F. Standardizing Melanopic Effects of Ocular Light for Ecological Lighting Design of Nonresidential Buildings—An Overview of Current Legislation and Accompanying Scientific Studies. Sustainability 2021, 13, 5131. https://doi.org/10.3390/su13095131

While melanopsin has been established as a primary contributor to continuous light, it is unknown whether it has any role in mediating responses to flash sequences. All of our exposures have a constant melanopic content, so this does not change the interpretation of the results. Our prior work indicates that it is unlikely that melanopsin plays a significant role in responses to flashed light.

There is certainly nothing wrong with what you write, but since the Manchester workshop in 2015, wavelength sensitive research has been heavily followed. Your content is relevant here and you should not leave out the topic completely. The information I queried may be of interest to colleagues. Add the details to the text or otherwise provide feedback.

Line 173. The maximum sensitivity of the human melatonin level to ambient light is achieved with light of a wavelength of more or less 490 nm, please describe to what extent you achieve the sensitivity with your illuminant in the experiment (4,000 K). Please tell the reader the reasons for the selection of the illuminant.

The 4000 K color temperature was selected as it provided a broad spectrum of light that could impact melanopsin, rods, and cones. It was not selected to preferentially activate any single photoreceptive system. We have added verbiage to this extent.

Unfortunately, the information content of your statement is quite low. You have not addressed the main point at all.

Author Response

We have added additional verbiage to the Discussion section to contextualize these data with the melanopsin system. While we appreciate the reviewer's encouragement to extend these results, we are concerned that we can only speculate about the potential roles of melanopsin and cones given the nature of the experimental paradigm and we do not wish to overstep our data.